# Thinking Forward and Backward: Effective Backward Planning with Large Language Models

## Abstract

Large language models (LLMs) have exhibited remarkable reasoning and planning capabilities. Most prior work in this area has used LLMs to reason through steps from an initial to a goal state or criterion, thereby effectively reasoning in a *forward* direction. Nonetheless, many planning problems exhibit an inherent *asymmetry* such that planning *backward* from the goal is significantly easier — for example, if there are bottlenecks close to the goal. We take inspiration from this observation and demonstrate that this bias holds for LLM planning as well: planning performance in one direction correlates with the planning complexity of the problem in that direction. However, our experiments also reveal systematic biases which lead to poor planning in the backward direction. With this knowledge, we propose a backward planning algorithm for LLMs that first flips the problem and then plans forward in the flipped problem. This helps avoid the backward bias, generate more diverse candidate plans, and exploit asymmetries between the forward and backward directions in planning problems — we find that combining planning in both directions with self-verification improves the overall planning success rates by 4-24% in three planning domains. Code: anonymous.repo.

## 1 Introduction

Large Language Models (LLMs) are increasingly capable in reasoning tasks — they can perform commonsense reasoning in a broad set of contexts (Kojima et al., 2022), reason about abstract patterns in data (Mirchandani et al., 2023), and learn to reason from human feedback (Liang et al., 2024). Such capabilities also open up the possibility of LLMs performing long-horizon planning (Valmeekam et al., 2024), where the model needs to reason about how the initial state and final goal of the problem can be connected through intermediate steps. Most existing work has explored such problems by asking the model to reason in the *forward* direction, i.e., generating intermediate steps from the initial state to the final goal. However, in many planning problems, there is an inherent *asymmetry*: generating the correct last steps leading to the goal can be much easier than generating the correct steps from the beginning. This leads to the question: *can LLMs plan better if they also reason in the backward direction?*

As an example, consider a robot navigating in a room ( Fig. 1): if the final goal is the bedroom at the end of a long and narrow hallway, it is natural to connect the bedroom with the beginning of the hallway first in the plan, and then search for the path that connects the hallway to the initial state. In this example, there is a "bottleneck" that causes the asymmetry: the number of possibilities when planning backward from the goal is constrained by the bottleneck (hallway), while the possibilities fan out quickly when planning from the start. Such bottlenecks are ubiquitous in planning problems; for example, in proving mathematical theorems (Loveland, 2016), there may be many possible steps to start a proof of a theorem, but the final steps can be much more closely related to the theorem statement and thus easier to choose. In our experiments, we quantify this bottleneck effect by comparing the exact number of search steps used by common forward-search algorithm (e.g., Breadth-First Search) when applied in the forward and backward directions on the underlying graph of the planning problem ( Fig. 1).

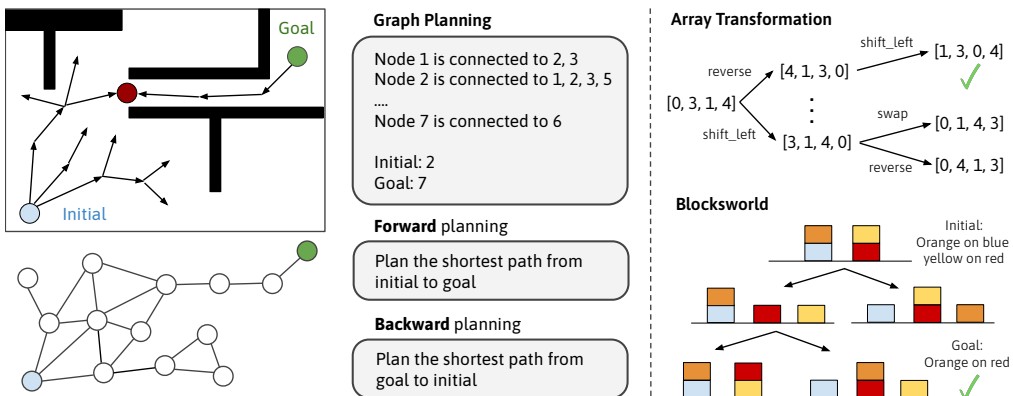

Figure 1: (Left) In planning problems such as navigation, it can often be easier to plan in the backward direction, especially when a "bottleneck" state (red node) exists and it is easier to find it from the goal rather than from the initial state. We consider GRAPH PLANNING, where the LLM needs to plan the shortest path from the initial node to the goal. (Right) Two other planning domains: ARRAY TRANSFORMATION, where an integer array is manipulated through different functions to the desired array, and BLOCKSWORLD, where stacks of blocks are re-oriented to the goal state.

The goals of this paper are twofold. First, we perform extensive experiments on three classical planning problems (Section 2) and examine whether LLM-based planning is significantly impacted by the asymmetry described above, i.e., if LLMs achieve higher planning success rates in the direction (forward/backward) where fewer computations are needed. Second, we study whether LLMs can exploit bottleneck effects by effectively planning backward (Section 3). Our results provide strong evidence for the first hypothesis. However, unfortunately, we find that LLMs exhibit a systematic bias of performing worse when planning backward; this may be attributed to the forward autoregressive nature of LLM output generation, as well as biases from the training dataset.

Addressing the backward reasoning bias, we propose a simple solution (Section 4): many problems can be transformed such that the original goal becomes the initial state and the original initial state becomes the goal — we ask LLMs to first "flip" the problem, and then plan in the forward direction (corresponding to the original backward direction). Given a planning problem, we ask LLMs to sample possible plans in the forward direction of both the original problem and the flipped one, and then self-verify (Stechly et al., 2024) all the plans before choosing the final one. We find that this simple setup allows LLMs to generate many more diverse candidate plans and exploit the bottleneck structure of the problems, improving the overall success rate in multiple planning domains by 4-24% (Section 5). Perhaps surprisingly, we also present evidence that in certain settings LLMs can also *reason whether to flip* the problem or not by examining the problem structure.

## 2 FORMULATION AND PLANNING DOMAINS

We define a (text-based) planning problem as $P = (\mathcal{S}, \mathcal{A_S}, s_0, g, T_{\max}, f)$, where $\mathcal{S}$ is the space of possible states, $\mathcal{A_S} = \{\mathcal{A}_s\}_{s \in \mathcal{S}}$ includes the space of possible actions (planned steps) at each state $s \in \mathcal{S}$, $s_0 \in \mathcal{S}$ is the initial state, $g \in \mathcal{S}$ is the goal, $T_{\max}$ is the maximum allowable steps, and $f$ is a ground-truth solution verifier that determines if a plan solves $P$, i.e., whether a plan connects $s_0$ to $g$, and possibly, is of optimal length. The generated plan $A = (a_0, ..., a_T)_{T \leq (T_{\max}-1)}$ is verified by $f$ based on these rules. Each action $a_t$ converts state $s_t$ to $s_{t+1}$, and we denote this as $s_t \xrightarrow{a_t} s_{t+1}$.

We denote pre-trained LLMs with parameters $\theta$ as $p_\theta$. Instead of the usual token-level generation, for convenience we consider LLMs generating the next step in a plan by sampling $a_t \sim p_\theta(a_t | s_0, g, \{a_0, ..., a_{t-1}\}, q)$, where $q$ denotes the rest of the text prompt (e.g., few-shot exemplars and instructions). For convenience, from now on we also assume that $q$ includes text descriptions of $s_0$, $g$, and $(a_0, ..., a_{t-1})$. During planning, LLMs may also generate other intermediaries such as the estimated states along the plan $(\hat{s}_1, ..., \hat{s}_t)$, which might help the LLM reason about the next step given the current estimated state.

Next we introduce the three different planning domains (Fig. 1) considered in our work. Appendix A provides more details on the design of these domains.

**Graph Planning.** In GRAPH PLANNING, each problem consists of a graph with $N$ nodes (labeled with a number $1, ..., N$). Each pair of nodes is connected with an edge with probability $\rho$; we consider both undirected and directed graphs. The initial state $s_0$ and goal $g$ are two different nodes. In order to vary difficulty, randomly generated graphs are filtered such that the shortest path from $s_0$ to $g$ has length $K$. In this domain, we consider a plan correct if it is optimal, i.e., shortest among paths between $s_0$ and $g$. We use a text-based *incident representation* for graphs (e.g., "Node 1 is connected to 2, 3"), which has been shown to be effective in the context of LLMs solving graph problems (Fatemi et al., 2023).

**Array Transformation.** In ARRAY TRANSFORMATION, the LLM is asked to convert an initial array of integers of length $N$ to a goal array, e.g., from $[0, 1, 3]$ to $[1, 0, 3, 1, 0, 3]$, using a set of array functions. We consider functions including: `repeat` to repeat the array once, `cut` to cut the array in half if the first half matches the second half, `shift_left` and `shift_right` to shift the array to the left by one (e.g., $[0, 1, 3] \rightarrow [1, 3, 0]$) or to the right, `reverse` to reverse the array, and `swap` to swap the first and last elements. Notice that each function has an inverse. We generate the problems by randomly sampling an initial array and a set of $K$ functions which are applied to obtain the final array. We consider a plan correct if it is valid, i.e., if planned functions convert the initial array to the goal. This domain can also be seen as a form of probabilistic context-free grammar (PCFG) (Sipser, 1996).

**Blocksworld.** BLOCKSWORLD is part of the PlanBench benchmark introduced by Valmeekam et al. (2024) for examining the overall planning capabilities of LLMs. In each problem, there are four blocks colored red, yellow, blue, or orange. The initial and goal states involve different possible stacks of the blocks, and the possible actions include `unstack <color> block from <color> block`, `pick up <color> block (from table)`, `put <color> block on <table>/<color> block`. We consider a plan correct if it is valid, i.e., if the planned steps move the initial stack to the goal.

## 3 LLMs Cannot Plan Backward As Effectively As Forward

As described in Section 1, many planning problems have an asymmetry that makes a given direction (forward vs. backward) easier for classical planning algorithms such as breadth-first search (BFS). Here, we examine whether the planning performance of LLMs is similarly impacted, and if the LLM's performance in a given planning direction can be predicted by the number of search steps required by BFS in that direction. If this is the case, it opens up the possibility of improving LLM-based planning by planing backward when the backward direction is favorable. Below we first introduce the backward planning algorithm we use for LLMs before showing the experimental results.

### 3.1 Backward planning

During backward planning, the LLM is given the same initial and goal states, but the prompt $q$ asks it to generate the plan $A$ in the reversed order. For example, consider an instance of GRAPH PLANNING where the shortest path from Node 1 to Node 3 is $(1, 5, 2, 3)$; then, the LLM should first output the backward plan as $(3, 2, 5, 1)$ and reverse the order before it proposes the plan to the verifier.

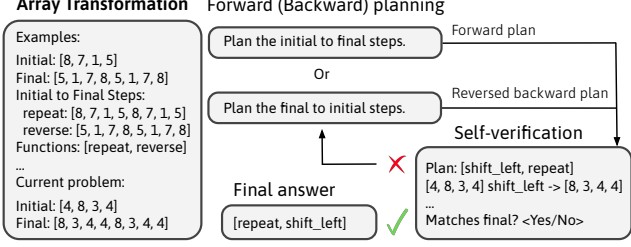

Figure 2: Sampling in forward or backward direction and then self-verifying the plans in ARRAY TRANSFORMATION.

As described in Section 1, backward planning can be useful when the search space from the goal is smaller than from the initial state. We can quantify the asymmetry between forward and backward planning by computing the number of search steps used by algorithms such as BFS in either direction.

### 3.2 ALGORITHM: SAMPLE FORWARD (BACKWARD) THEN SELF-VERIFY

While the basic version of LLM planning involves asking it to generate a single plan, we consider a more robust version where the LLM first samples multiple candidate solutions in one direction, and then self-verifies them to choose the final solution; then the verifier $f$ checks the final plan. In both stages, the LLM is shown few-shot exemplars to familiarize itself with the problem and to learn to verify the candidate plans step by step. The overall algorithm is illustrated in Fig. 2. Given the maximum number of attempts $M$, the sampling temperature of the LLM is set to be 0 for the first attempt, and nonzero for the later ones. Self-verification works differently depending on the domain: in GRAPH PLANNING, since the solution needs to be optimal, we save all the unique candidate plans $\{A_j\}$ after all $M$ attempts, and then present all of them to the LLM and ask it to verify them as well as to choose the optimal one; in ARRAY TRANSFORMATION and BLOCKSWORLD, since the solution does not need to be optimal, the LLM verifies each candidate plan as soon as it is sampled, and either stops when it deems one plan correct or after all attempts. For all the experiments, we use the `GPT-4o` model from OpenAI unless noted otherwise.

### 3.3 RESULTS

**LLM plans better in the easier direction.** Here we run experiments with both directed and undirected graphs in GRAPH PLANNING, and ask the LLM to plan either forward or backward. We also compute the number of search steps used by BFS in both directions. Fig. 3 shows the planning success rates achieved by forward and backward planning at different levels of forward/backward difficulty (as quantified by BFS computations). For either direction, the success rate is generally higher when the number of computations is lower in that direction. This finding suggests that LLM planning is akin to a forward-search algorithm such as BFS in terms of the difficulty of planning in a given direction, and thus can be potentially improved by planning backward when it is easier. We also find similar results with ARRAY TRANSFORMATION shown in Appendix B.

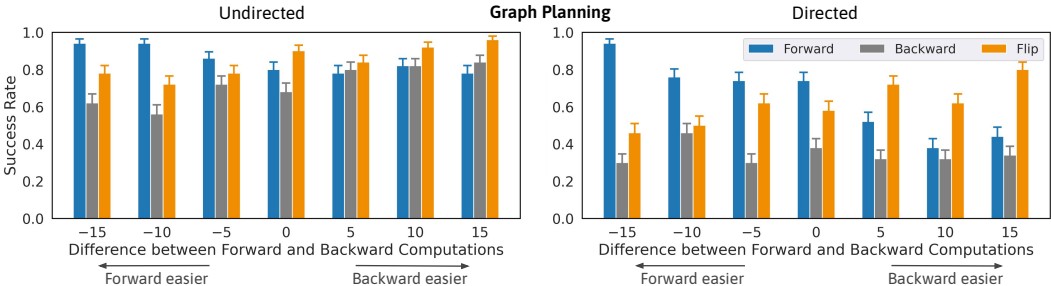

Figure 3: Success rates achieved by forward and backward planning in GRAPH PLANNING vs. difference between forward and backward BFS computations. The LLM plans better in the direction of fewer computations needed, but the forward direction outperforms backward in general.

**LLM plans worse in the backward direction.** Next, we study the effectiveness of backward planning by examining whether the LLM can achieve the same level of planning success in the backward direction as compared to forward. We find that this is not the case. Fig. 3 shows that the backward success rate is consistently lower than forward. We also calculate the average planning success rates for all four settings (from experiments in Section 5) —

| Domain | Forward | Backward |
|---|---|---|
| Graph Planning (undirected) | $82.5_{\pm2.7}\%$ | $76.7_{\pm3.0}\%$ |
| Graph Planning (directed) | $38.5_{\pm3.4}\%$ | $29.7_{\pm3.2}\%$ |
| Array Transformation | $67.5_{\pm3.3}\%$ | $62.2_{\pm3.4}\%$ |
| Blocksworld | $39.5_{\pm3.5}\%$ | $20.5_{\pm2.9}\%$ |

Table 1: Average planning success rate in forward vs. backward direction in each setting.

as shown in Table 1, the LLM consistently plans worse in the backward direction. We conjecture that this bias may be attributed to the forward (i.e., left to right) autoregressive nature of LLM output generation, as well as biases from the training dataset. Next we propose a solution to such backward bias and allow LLMs to plan effectively in the backward direction (*Flip* in Fig. 3).

## 4 PROPOSED SOLUTION: PLAN BACKWARD BY FLIPPING THE PROBLEM

If the LLM cannot plan backward as well as forward, how can it effectively exploit the backward direction in planning? We propose a simple solution: *flip* the problem such that the original goal becomes the new initial state and vice versa. The LLM can then plan in the forward direction for the flipped problem, which corresponds to the backward direction of the original problem. This avoids the bias of weak LLM planning in the backward direction. However, there are a few subtleties that must be taken care of when flipping the planning problem.

**Change of the state-dependent action space.**   In some cases, $\mathcal{A}_{\mathcal{S}}$ needs to be adjusted for the flipped problem. For example, with a directed graph, an edge from Node 1 to Node 3 in the original problem corresponds to an edge from Node 3 to Node 1 in the flipped problem — Node 3 might not be reachable from Node 1 in the flipped problem. In contrast, there is no change needed for undirected graphs, ARRAY TRANSFORMATION, and BLOCKSWORLD. With directed graphs, we prompt the LLM to first generate the new text representation of the graph (Fig. 4 right), and then generate a plan for the flipped problem.

**Flipping back the plan.**   After the plan $A'$ for the flipped problem is generated, steps within it need to be reversed in order and also often "flipped back" to generate a forward plan for the original problem:

$$\text{Flip back the plan: } A' = \{a'_0, ..., a'_T\} \mapsto A = \{a_T, ..., a_0\}, \text{where } s_t \xrightarrow{a'_t} s_{t+1}, s_{t+1} \xrightarrow{a_t} s_t. \quad (1)$$

We assume that each action $a \in \mathcal{A}_s$ can be inverted, i.e., if $s \xrightarrow{a} s'$, there exists $a' \in \mathcal{A}_{s'}$ such that $s' \xrightarrow{a'} s$. In practice, we prompt the LLM to generate the corresponding $a_t$ after it generates $a'_t$. Fig. 4 left shows an example for BLOCKSWORLD: the action "put yellow on red" is flipped back to "unstack yellow from red", after the order of the plan is flipped.

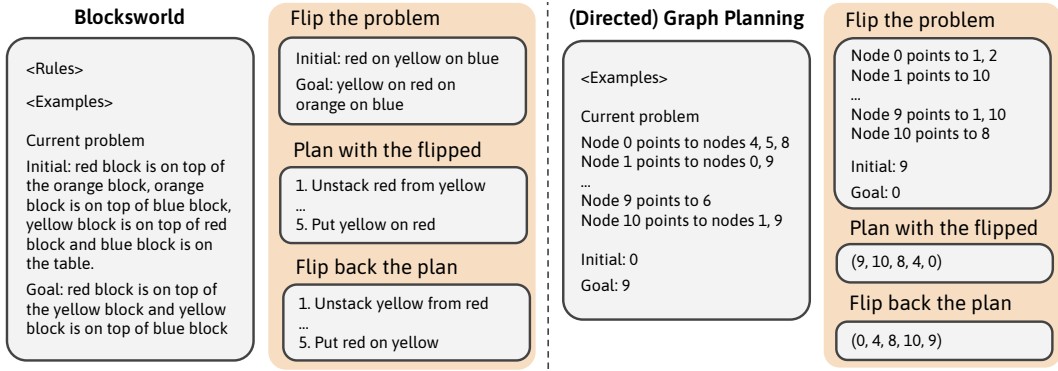

Figure 4: Flipping the problem and then plan in the new forward direction in BLOCKSWORLD (left) and GRAPH PLANNING with directed graph (Right).

**Algorithm: Sample forward and flipped plans, then self-verify.**   Extending the algorithm from Section 3.2 where multiple plans in the forward or backward directions are sampled before self-verifying them, we can now sample in the forward direction of the flipped problem, instead of sampling backward in the original problem. Sampling forward with the original *and the flipped* problem generates more diverse solutions for self-verification, while avoiding the backward bias and potentially exploiting asymmetries. We can either randomly sample the problem (original or flipped) at each attempt, or prompt the LLM to reason which direction is favorable.

## 5 EXPERIMENTS WITH FLIPPING THE PROBLEM

With experiments in the three planning domains introduced in Section 2, we investigate the following questions in the corresponding subsections below:

Q1: Does flipping the problem help improve the planning success rate?

Q2: When does flipping the problem help the most?

Q3: How well can the LLM reason about when to flip the problem?

**Baselines.** We compare a few different ways to sample candidate solutions before self-verification: (1) **Fwd**: only planning in the forward direction; (2) **Back**: only planning in the backward direction; (3) **Flip**: only planning in the forward direction of the flipped problem; (4) **Fwd-Back**: randomly choose either forward or backward direction and then plan; (5) **Fwd-Flip**: randomly choose either forward direction of the original problem or forward direction of the flipped problem and then plan; (6) **Fwd-Flip-Reason**: have the LLM reason whether to plan forward in the original or flipped problem and then plan. We expect our proposed **Fwd-Flip** to outperform Fwd, Back, Flip, Fwd-Flip.

### 5.1 BACKWARD PLANNING WITH THE FLIPPED PROBLEM HELPS IMPROVE SUCCESS RATE

In order to test whether flipping the problem can help planning success rate, we run extensive experiments with the three planning domains and compare Fwd, Back, Flip, Fwd-Back, and Fwd-Flip. We use a maximum $M = 6$ attempts for all experiments. Table 2 shows each result averaged over 200 trials. Flip outperforms Back and matches Fwd performance; we also see the similar trends in Fig. 3. In addition, Fwd-Flip consistently leads to the highest planning success rate, improving by 4-24% over Fwd, except for one setting where all baselines achieve close to 100% success. These results corroborate that flipping the problem mitigates the backward bias.

| Domain | | | Fwd | Back | Flip | Fwd-Back | Fwd-Flip |
|---|---|---|---|---|---|---|---|
| | Directed? | $\rho$ | $N$ | | | | |
| | No | 0.2 | 12 | $82.5_{\pm2.7}\%$ | $74.5_{\pm3.1}\%$ | $84.5_{\pm2.6}\%$ | $91.0_{\pm2.0}\%$ | $\mathbf{93.5_{\pm1.7}\%}$ |
| | No | 0.2 | 10 | $89.5_{\pm2.2}\%$ | $78.0_{\pm2.9}\%$ | $88.0_{\pm2.3}\%$ | $93.0_{\pm1.8}\%$ | $\mathbf{96.5_{\pm1.3}\%}$ |
| Graph Planning | No | 0.3 | 10 | $90.5_{\pm2.1}\%$ | $77.0_{\pm3.0}\%$ | $91.0_{\pm2.0}\%$ | $93.0_{\pm1.8}\%$ | $\mathbf{94.5_{\pm1.6}\%}$ |
| | Yes | 0.2 | 12 | $62.5_{\pm3.4}\%$ | $29.0_{\pm3.2}\%$ | $67.0_{\pm3.3}\%$ | $76.0_{\pm3.0}\%$ | $\mathbf{80.0_{\pm2.8}\%}$ |
| | Yes | 0.2 | 10 | $73.0_{\pm3.1}\%$ | $36.5_{\pm3.4}\%$ | $73.5_{\pm3.1}\%$ | $79.5_{\pm2.9}\%$ | $\mathbf{88.5_{\pm2.3}\%}$ |
| | Yes | 0.3 | 10 | $69.5_{\pm3.3}\%$ | $23.5_{\pm3.0}\%$ | $64.0_{\pm3.4}\%$ | $68.5_{\pm3.3}\%$ | $\mathbf{86.5_{\pm2.4}\%}$ |
| | Functions | | | | | | | |
| | `shift, repeat, cut` | | $99.0_{\pm0.7}\%$ | $98.5_{\pm00.9}\%$ | $99.0_{\pm0.7}\%$ | $\mathbf{100.0_{\pm0.0}\%}$ | $99.5_{\pm0.5}\%$ |
| Array Transformation | `shift, reverse, swap` | | $50.0_{\pm3.5}\%$ | $36.0_{\pm3.4}\%$ | $52.0_{\pm3.5}\%$ | $46.0_{\pm3.5}\%$ | $\mathbf{56.0_{\pm3.5}\%}$ |
| | `repeat, cut, reverse, swap` | | $53.5_{\pm3.5}\%$ | $52.0_{\pm3.5}\%$ | $53.0_{\pm3.5}\%$ | $54.5_{\pm3.5}\%$ | $\mathbf{56.5_{\pm3.5}\%}$ |
| Blocksworld | - | | | $39.5_{\pm3.5}\%$ | $20.5_{\pm2.9}\%$ | $34.5_{\pm3.4}\%$ | $27.0_{\pm3.1}\%$ | $\mathbf{48.5_{\pm3.5}\%}$ |

Table 2: Planning success rate averaged over 200 trials for the five methods in the three planning domains. Flip matches Fwd. Fwd-Flip generally achieves the highest success rate.

**Fwd-Flip exploits asymmetries in the problems.** One of the main motivations of planning backward is that many planning problems have a bottleneck structure: there exists a less connected part of the state space near the initial state or the goal — this makes it easier to plan from the end closer to the bottleneck. In Fig. 5, we calculate the difference between forward vs. backward computations for BFS in GRAPH PLANNING, bin them, and find the success rate of each bin for Fwd and Fwd-Flip — the plot shows the average success rate over the three directed graph settings (shades show the minimum and maximum over the three). We find that Fwd tends to perform worse when the forward computations needed are higher, meaning that it cannot plan

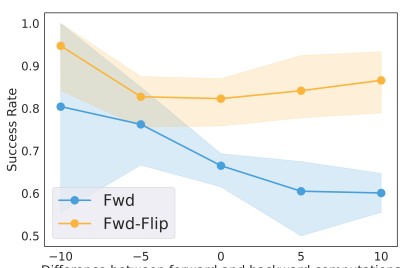

Figure 5: Fwd-Flip plans well even when the forward direction is difficult.

as effectively when the forward direction is more difficult. In contrast, Fwd-Flip maintains a similar level of success regardless of forward vs. backward planning difficulty.

**Flipping the problem generates more diverse candidate solutions.** The improved performance of Fwd-Flip over Fwd can also be partly attributed to the generation of more diverse solutions. Fig. 6

compares Fwd vs. Fwd-Back vs. Fwd-Flip in ARRAY TRANSFORMATION and shows both the average number of unique candidate solutions generated (top) and the planning success rate (bottom) vs. the number of planning attempts, $M$. Fwd-Flip generates a higher number of unique candidates, and improves the planning success rate by a large margin. Fig. 7 (left) shows an example where Fwd fails to solve the block task due to a persistent error (trying to move a block before "unstacking" it, which is not allowed by the rules of BLOCKSWORLD) even with significant sampling temperature (0.4), while Fwd-Flip generates a different solution and avoids the error.

## 5.2 IMPROVEMENT FROM FLIPPING THE PROBLEM VARIES

**Effect of backward success rate without flipping.** While in Section 5.1 we have shown that flipping the problem helps the LLM exploit asymmetries between forward and backward planning, we tend to find that the performance difference between Fwd-Flip and Fwd-Back varies substantially in ARRAY TRANSFORMATION (Table 2). When functions `repeat` and `cut` are used, Back does not perform significantly worse than Fwd — we believe this is due to the backward step being easily recognizable by Back (Fig. 7 right) when the initial and goal array sizes differ (since `repeat` and `cut` change them). In this case, Fwd-Flip does not improve much over Fwd-Back. With functions `shift_left|right`, `reverse`, `swap` that do not change array sizes, we see a much more significant backward bias (53.5% vs. 36% between Fwd and Back) and there is a bigger margin between Fwd-Flip and Fwd-Back (56% vs. 46%).

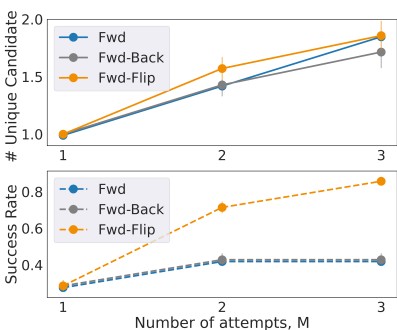

Figure 6: Fwd-Flip generates diverse solutions and improves success rate.

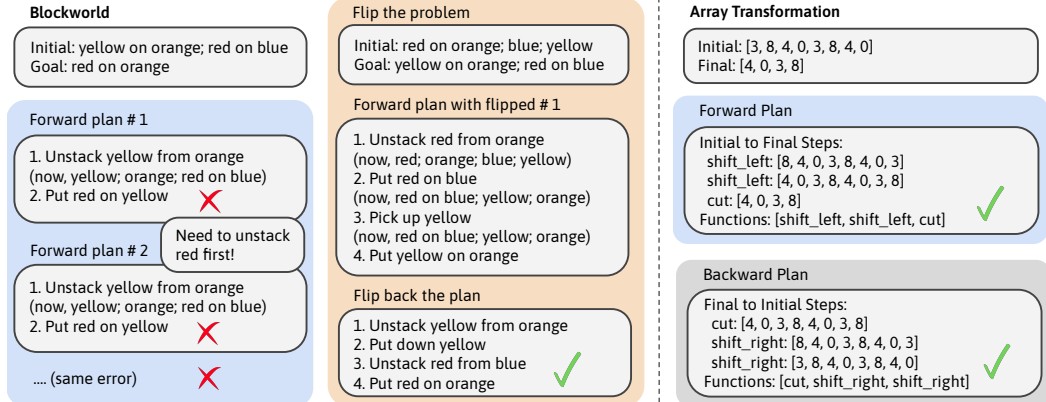

Figure 7: (Left) Flipping the problem helps LLM avoid persistent error in one direction. (Right) With functions `cut` and `repeat` used in ARRAY TRANSFORMATION that change the array size, it can become obvious that one of them has to be used, which makes backward planning easier.

**Effect of LLM capability.** We also hypothesize that the effect of planning in the flipped problem also depends on the inherent capability of the LLM. First, regardless of the choice of the LLM, Fwd-Flip should consistently improve the success rate. In Fig. 8 we run `GPT-3.5-turbo` and `GPT-4-turbo` besides `GPT-4o` with a directed graph setting and in BLOCKSWORLD, and we see that Fwd-Flip consistently outperforms Fwd. We also find that the design choices of the algorithm can affect the performance, and we highlight two aspects below:

- Reliability of self-verification: while we allow the LLM to generate multiple candidates (in either direction), it can only present a single solution after it self-verifies the candidates. Hence, the LLM needs to reliably self-verify the candidate plans to achieve a high success rate. In addition to the success rate, Fig. 8 shows the self-verification error rate. We find that the self-verification error reduces as the LLM becomes more capable. In BLOCKSWORLD, we find that `GPT-3.5-turbo` always self-verifies its sampled plan to be correct, leading to close to zero success rate.

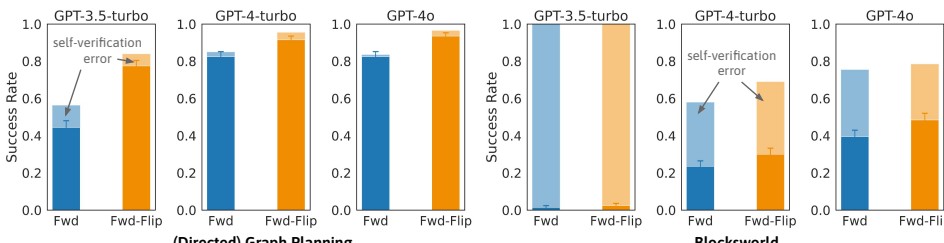

Figure 8: Planning performance of different LLMs with Fwd and Fwd-Flip in (directed) GRAPH PLANNING and BLOCKSWORLD. Flipping the problem helps all LLMs plan better, and better LLMs reduce errors when executing the algorithm.

- Reliability of flipping the state-dependent action space: in Section 4 we described how flipping the problem requires flipping back the plan and possibly changing the state-dependent action space. While we find that the LLM can flip back the plan itself without error, sometimes it can be error-prone when flipping the action space, affecting the performance of Flip and Fwd-Flip. In GRAPH PLANNING with directed graphs, flipping the problem involves flipping the edges and re-ordering them — Fig. 9 shows an example. Possible errors in re-ordering the edges cause Flip to under-perform Fwd in some of the directed graph settings, while Flip always matches Fwd with undirected graphs.

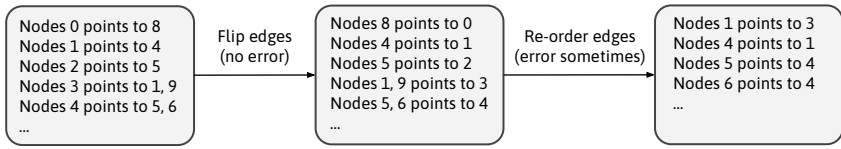

Figure 9: When flipping the directed graph, we find it is helpful to re-order the flipped edges such that they appear in the order of "Node 0", "Node 1", etc., to achieve better success rate. However, this also introduces possible error, causing Flip to under-perform Fwd sometimes and affecting Fwd-Flip.

Nonetheless, as Fig. 8 illustrates, these errors are reduced with better LLMs. Flipping the problem improves success rates significantly even with improvements in the underlying LLM, demonstrating the scalability of our method.

**Effect of initial/goal asymmetry.** We also notice that in BLOCKSWORLD, Flip under-performs Fwd (Table 2). The reason is that while the initial states of the problems are completely specified (e.g., "yellow block on red block, orange block on blue block"), the goal can be partially specified (e.g., "red block on orange block"). This means that when the initial and goal states are flipped, the LLM needs to first generate a complete flipped initial state from the original goal state — planning with full initial and goal states can be more challenging than with full initial and partial goal states.

**Effect of the number of planning attempts.** Lastly, we also find that the maximum number of planning attempts, $M$, can affect the relative performance between Fwd and Fwd-Flip — this is particularly true when Flip under-performs Fwd such as in BLOCKSWORLD due to the inital/goal asymmetry mentioned above. In Fig. 10 we show Fwd vs. Fwd-Flip as $M$ varies from 1 to 6 — Fwd-Flip under-performs Fwd slightly at $M = 1$, but out-performs Fwd with $M > 1$ with increasing improvements.

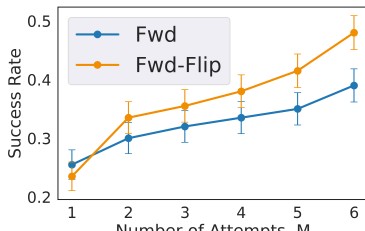

Figure 10: Fwd-Flip performs better given more planning attempts.

### 5.3 LLM CAN REASON WHEN TO FLIP IN CERTAIN SETTINGS

With Fwd-Flip we randomly choose between the original and the flipped problem for each candidate solution. We now explore whether the LLM can also reason when to flip the problem — we

Figure 11: LLM can analyze the graph and choose the easier direction for planning in zero-shot.

hypothesize that the LLM can analyze the problem structure to some extent and reason whether the backward direction can be easier. We run experiments with one setting from each of the planning domains (Table 3); unlike previous experiments, here we use $M = 1$ as we find that the LLM is often certain about the direction. We find that Fwd-Flip-Reason out-performs Fwd and Flip in undirected graphs. Fig. 11 shows an example where the LLM identifies the bottleneck near the goal and chooses to plan backward, leading to the correct answer. We then calculate the rate at which the LLM chooses the direction where the number of computations is lower and find that the LLM can choose the easier direction 78.5% of the time with undirected graphs and 60.5% of the time with directed graphs. However, with directed graphs, the LLM suffers from possible errors when flipping the problem, and thus Fwd-Flip-Reason with $M = 1$ does not perform better than Fwd.

In ARRAY TRANSFORMATION and BLOCKSWORLD, since the problem asymmetry is less clear, we do not expect the LLM to find a better direction in general, and thus Fwd-Flip-Reason does not out-perform Fwd or Flip. Appendix A provides details on the prompts used to elicit LLM reasoning.

| Domain | Fwd | Flip | Fwd-Flip-Reason |
|---|---|---|---|
| Graph Planning (Undirected) | $79.0_{\pm 2.9}\%$ | $78.5_{\pm 2.9}\%$ | $\mathbf{87.5_{\pm 2.3}\%}$ |
| Graph Planning (Directed) | $\mathbf{50.5_{\pm 3.5}\%}$ | $46.0_{\pm 3.5}\%$ | $\mathbf{50.5_{\pm 3.5}\%}$ |
| Array transformation (`shift`, `reverse`, `swap`) | $18.5_{\pm 2.7}\%$ | $\mathbf{19.5_{\pm 2.8}\%}$ | $16.5_{\pm 2.6}\%$ |
| Blocksworld | $\mathbf{33.0_{\pm 3.3}\%}$ | $21.5_{\pm 2.9}\%$ | $27.0_{\pm 3.1}\%$ |

Table 3: Planning success rate averaged over 200 trials for the five methods in the three planning domains when the LLM is asked to choose the direction to reason in. The maximum planning attempt $M$ is set to 1.

## 6 RELATED WORK

**Bidirectional planning.** It is well known in classical planning (LaValle, 2006) that searching and planning from the backward direction can often reduce the computations needed. Bi-directional search has been incorporated into popular sampling-based planners (e.g., rapidly-exploring random trees) (Jordan & Perez, 2013) and heuristics-based techniques (e.g., A*) (Kuroiwa & Fukunaga, 2020) to improve efficiency.

**Planning with LLMs.** LLMs have been widely applied in different planning problems (Kambhampati et al.) such as text games (Yao et al., 2024; Wang et al., 2023), robot planning (Huang et al., 2022; Ahn et al., 2022; Ren et al., 2023), scientific experimentation (Wang et al., 2022a), and web navigation (Deng et al., 2024) — their strong planning capabilities originate from pre-training with enormous amounts of text data, which elicits reasoning capabilities (Wei et al., 2022). Beyond simple zero- or few-shot learning, LLMs have also been combined with classical planners (Liu et al., 2023; Silver et al., 2024) and external tools (Zeng et al., 2022) to further boost performance. However, no

previous work has systematically studied asymmetries between forward and backward planning for LLMs, or leveraged backward reasoning to improve LLM planning.

**Backward reasoning in LLMs.**    Despite not being applied in planning problems, backward reasoning has been used with LLMs for self-verifying forward-sampled solutions to math problems (Jiang et al., 2023). Another line of work (Kazemi et al., 2022; Lee & Hwang, 2024) applies backward chaining, which recursively breaks the target into sub-targets based on defined rules, mostly for verifying a statement or proof. Our work instead proposes a general solution to planning problems using both forward and backward reasoning, and is motivated by a systematic examination of asymmetries in forward and backward planning for LLMs.

## 7 CONCLUSION AND FUTURE WORK

In this work, we investigate how to enable LLMs to effectively plan in the backward direction from the desired goal to improve the overall planning success rate. First, our experiments reveal consistently worse performance of LLMs when planning backward. To address this, we propose instructing the LLM to flip the problem first and then plan forward in the flipped problem. Combined with self-verification, we find that generating candidate solutions from both directions improves planning success rates by 4-24% over forward-only planning in three different domains.

One immediate future direction is to better teach LLMs to reason and plan backward, e.g., by fine-tuning with correct forward and *backward* reasoning traces (Zelikman et al., 2022). We also believe that our framework of combining forward and backward reasoning can be extended to general reasoning problems, e.g., allowing LLMs to generate more diverse reasoning traces from the backward direction (Yao et al., 2024), or, enforcing reasoning self-consistency from both forward and backward directions (Wang et al., 2022b).

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

## A APPENDIX - PLANNING DOMAINS

We provide all codes needed to replicate the experiments in the attached supplementary materials.

### A.1 GRAPH PLANNING

**Configurations.** To generate the desired graphs, we use `gnp_random_graph` function from the `networkx` package, with which we specify $N$, the number of nodes, $\rho$, the probability of two nodes are connected with an edge, and whether the edges are directed or not. We also apply rejection sampling to ensure the shortest path involves a total of five nodes. The the initial and goal nodes are sampled randomly from the graph.

**Prompts.** For both the prompts for sampling the candidate plans and self-verifying them, we randomly generate three similar graphs as few-shot examples (listing 1, listing 2). With directed graph, we also prompt the model to re-order the flipped edges such that the graph still starts with "Node 0 points to ..." instead of "Node 7, 9 points to 0", if the original problem has "Node 0 points to 7, 9." ( listing 3), which we find improve the planning success with the flipped problem. listing 4 shows the prompt used for eliciting LLM's preference over the planning directions in zero-shot. We use temperature $T = 0$ for re-ordering, self-verifying, eliciting preference, and the first planning attempt, and $T = 0.5$ for the rest of planning attempts.

Listing 1: Sample prompt used in GRAPH PLANNING for sampling candidate solution. We use "is connected to" for edges in undirected graphs, and "points to" for edges in directed graph. For backward planning, we use "Plan the shortest path from goal to initial node..." as the final instruction.

```
You will be given an undirected graph search problem with a few examples.

** Example 1 **
Node 1 is connected to nodes 2, 3, 6, 11
Node 2 is connected to nodes 1, 3
Node 3 is connected to nodes 1, 2
Node 4 is connected to nodes 7
Node 6 is connected to nodes 1, 9, 10
Node 7 is connected to nodes 4, 8, 9
Node 8 is connected to nodes 7, 11
Node 9 is connected to nodes 6, 7
Node 10 is connected to nodes 6, 11
Node 11 is connected to nodes 1, 8, 10

Initial: 3
Goal: 7
Shortest Path: (3, 1, 11, 8, 7)

** Example 2 **
Node 0 is connected to nodes 3, 4, 6, 7, 9
Node 1 is connected to nodes 10, 11
Node 2 is connected to nodes 7
Node 3 is connected to nodes 0, 9, 11
Node 4 is connected to nodes 0
Node 5 is connected to nodes 10, 11
Node 6 is connected to nodes 0
Node 7 is connected to nodes 0, 2, 8, 9, 11
Node 8 is connected to nodes 7, 11
Node 9 is connected to nodes 0, 3, 7
Node 10 is connected to nodes 1, 5, 11
Node 11 is connected to nodes 1, 3, 5, 7, 8, 10

Initial: 6
Goal: 1
Shortest Path: (6, 0, 7, 11, 1)

** Example 3 **
Node 0 is connected to nodes 1, 9
Node 1 is connected to nodes 0, 3, 5
Node 2 is connected to nodes 3, 7, 10
Node 3 is connected to nodes 1, 2, 6, 11
Node 4 is connected to nodes 8
Node 5 is connected to nodes 1, 7, 11
Node 6 is connected to nodes 3, 11
Node 7 is connected to nodes 2, 5
Node 8 is connected to nodes 4, 9
Node 9 is connected to nodes 0, 8
Node 10 is connected to nodes 2
Node 11 is connected to nodes 3, 5, 6

Initial: 8
Goal: 3
Shortest Path: (8, 9, 0, 1, 3)
```

```
** Current problem **
Node 0 is connected to nodes 5
Node 1 is connected to nodes 2, 4, 9
Node 2 is connected to nodes 1, 5, 7
Node 3 is connected to nodes 6, 7
Node 4 is connected to nodes 1
Node 5 is connected to nodes 0, 2, 6, 11
Node 6 is connected to nodes 3, 5
Node 7 is connected to nodes 2, 3, 10
Node 8 is connected to nodes 9
Node 9 is connected to nodes 1, 8
Node 10 is connected to nodes 7
Node 11 is connected to nodes 5

Initial: 6
Goal: 4

Plan the shortest path from initial to goal node for the this **undirected** graph. Follow the format 'Shorest
        Path: (...)' and do not output anything else.
```

Listing 2: Sample prompt used in GRAPH PLANNING for self-verifying the candidate solutions.

```
You will be given a directed graph search problem with a few examples.

** Example 1 **
Node 0 points to nodes 1, 8, 10, 11
Node 1 points to nodes 0, 3, 4
Node 2 points to nodes 3
Node 3 points to nodes 4, 6
Node 4 points to nodes 2, 5, 7, 9
Node 5 points to nodes 9
Node 6 points to nodes 0, 7, 11
Node 7 points to nodes 5, 10
Node 8 points to nodes 0, 11
Node 9 points to nodes 7, 11
Node 10 points to nodes 4, 7, 8, 9
Node 11 points to nodes

Initial: 6
Goal: 9

Which one is the correct shortest path?
A. (6, 0, 1, 4, 7, 10, 9)
B. (6, 7, 5, 9)
Checking each options step by step:
A: check 6 to 0, 6 points to [0, 7, 11], 0 in [0, 7, 11]? True; check 0 to 1, 0 points to [1, 8, 10, 11], 1 in
        [1, 8, 10, 11]? True; check 1 to 4, 1 points to [0, 3, 4], 4 in [0, 3, 4]? True; check 4 to 7, 4 points
        to [2, 5, 7, 9], 7 in [2, 5, 7, 9]? True; check 7 to 10, 7 points to [5, 10], 10 in [5, 10]? True;
        check 10 to 9, 10 points to [4, 7, 8, 9], 9 in [4, 7, 8, 9]? True – valid path of length 7
B: check 6 to 7, 6 points to [0, 7, 11], 7 in [0, 7, 11]? True; check 7 to 5, 7 points to [5, 10], 5 in [5,
        10]? True; check 5 to 9, 5 points to [9], 9 in [9]? True – valid path of length 4
Valid options: A with length length 7, B with length length 4. Thus the correct shortest option is B

** Example 2 **
Node 0 points to nodes 7, 8, 11
Node 1 points to nodes 3, 4, 5, 10
Node 2 points to nodes
Node 3 points to nodes 4, 8, 9
Node 4 points to nodes 3, 8
Node 5 points to nodes 1, 8, 11
Node 6 points to nodes 0, 1, 3, 11
Node 7 points to nodes 1, 2, 6, 9, 11
Node 8 points to nodes 5
Node 9 points to nodes 6
Node 10 points to nodes 0, 2, 5, 6, 7
Node 11 points to nodes 0, 2, 3, 4, 9

Initial: 7
Goal: 11

Which one is the correct shortest path?
A. (7, 6, 10, 0, 8, 5, 11)
B. (7, 11)
C. (7, 1, 9, 6, 0, 11)
Checking each options step by step:
A: check 7 to 6, 7 points to [1, 2, 6, 9, 11], 6 in [1, 2, 6, 9, 11]? True; check 6 to 10, 6 points to [0, 1,
        3, 11], 10 in [0, 1, 3, 11]? False – invalid path
B: check 7 to 11, 7 points to [1, 2, 6, 9, 11], 11 in [1, 2, 6, 9, 11]? True – valid path of length 2
C: check 7 to 1, 7 points to [1, 2, 6, 9, 11], 1 in [1, 2, 6, 9, 11]? True; check 1 to 9, 1 points to [3, 4,
        5, 10], 9 in [3, 4, 5, 10]? False – invalid path
Valid options: B with length length 2. Thus the correct shortest option is B

** Example 3 **
Node 0 points to nodes 2, 4, 8, 9
Node 1 points to nodes 3, 7, 8, 10
Node 2 points to nodes 8, 10
Node 3 points to nodes 1, 4, 8, 9, 10
Node 4 points to nodes 7, 9
Node 5 points to nodes 3, 7
```

```
Node 6 points to nodes 5, 11
Node 7 points to nodes 2, 10
Node 8 points to nodes 0, 3, 7
Node 9 points to nodes 10
Node 10 points to nodes 0, 4
Node 11 points to nodes 3, 7

Initial: 2
Goal: 0

Which one is the correct shortest path?
A. (2, 10, 0)
B. (2, 8, 3, 9, 7, 0)
C. (2, 10, 0)
D. (2, 10, 3, 1, 10, 0)
Checking each options step by step:
A: check 2 to 10, 2 points to [8, 10], 10 in [8, 10]? True; check 10 to 0, 10 points to [0, 4], 0 in [0, 4]?
    True - valid path of length 3
B: check 2 to 8, 2 points to [8, 10], 8 in [8, 10]? True; check 8 to 3, 8 points to [0, 3, 7], 3 in [0, 3, 7]?
    True; check 3 to 9, 3 points to [1, 4, 8, 9, 10], 9 in [1, 4, 8, 9, 10]? True; check 9 to 7, 9 points
    to [10], 7 in [10]? False - invalid path
C: check 2 to 10, 2 points to [8, 10], 10 in [8, 10]? True; check 10 to 0, 10 points to [0, 4], 0 in [0, 4]?
    True - valid path of length 3
D: check 2 to 10, 2 points to [8, 10], 10 in [8, 10]? True; check 10 to 3, 10 points to [0, 4], 3 in [0, 4]?
    False - invalid path
Valid options: A with length length 3, C with length length 3. Thus the correct shortest option is C

** Current problem **
Node 0 points to nodes 2, 3, 4, 8, 10
Node 1 points to nodes 2, 6, 7
Node 2 points to nodes 4, 5
Node 3 points to nodes 2
Node 4 points to nodes 2, 5, 6, 9, 11
Node 6 points to nodes 2, 3, 4, 8
Node 7 points to nodes 9
Node 9 points to nodes 10, 11
Node 10 points to nodes 0, 8, 9
Node 11 points to nodes 1, 4, 9

Initial: 3
Goal: 1
Which one is the correct shortest path?
A. (3, 2, 4, 11, 1)
Remember the graph is directed. Follow the exact same format as the examples and check each options step by
    step. Begin with 'Checking each options step by step:'
```

Listing 3: Sample prompt used in GRAPH PLANNING for re-ordering the flipped directed graph. The example here is the same for all problems

```
You will be asked to re-order a directed graph.

** Example **
Nodes 8 points to node 0
Nodes 4, 10 points to node 1
Nodes 5 points to node 2
Nodes 1, 9, 11 points to node 3
Nodes 5, 6, 11 points to node 4
Nodes   points to node 5
Nodes 2, 9 points to node 6
Nodes 1, 10 points to node 7
Nodes 2, 4, 10 points to node 8
Nodes 10 points to node 9
Nodes 2, 3, 7 points to node 10
Nodes 1, 2 points to node 11

Full procedure:
1. List all directed edges
8 -> 0
4 -> 1
10 -> 1
5 -> 2
1 -> 3
9 -> 3
11 -> 3
5 -> 4
6 -> 4
11 -> 4
2 -> 6
9 -> 6
1 -> 7
10 -> 7
2 -> 8
4 -> 8
10 -> 8
10 -> 9
2 -> 10
3 -> 10
7 -> 10
1 -> 11
2 -> 11
```

```
2. Group the edges for each node
0 ->
1 -> 3, 7, 11
2 -> 6, 8, 10, 11
3 -> 10
4 -> 1, 8
5 -> 2, 4
6 -> 4
7 -> 10
9 -> 3, 6
10 -> 1, 7, 8, 9
11 -> 3, 4
3. Convert the edges into the text format
Node 1 points to nodes 3, 7, 11
Node 2 points to nodes 6, 8, 10, 11
Node 3 points to node 10
Node 4 points to nodes 1, 8
Node 5 points to nodes 2, 4
Node 6 points to node 4
Node 7 points to node 10
Node 8 points to node 0
Node 9 points to nodes 3, 6
Node 10 points to nodes 1, 7, 8, 9
Node 11 points to nodes 3, 4

** Current Graph **
Nodes 2, 3, 4, 8, 10 points to node 0
Nodes 2, 6, 7 points to node 1
Nodes 4, 5 points to node 2
Nodes 2 points to node 3
Nodes 2, 5, 6, 9, 11 points to node 4
Nodes 2, 3, 4, 8 points to node 6
Nodes 9 points to node 7
Nodes 10, 11 points to node 9
Nodes 0, 8, 9 points to node 10
Nodes 1, 4, 9 points to node 11

Remember the edges are directed. Please re-order this directed graph with the exact same full procedure as the
    example. Follow the same format and do not output anything else.
```

Listing 4: Sample prompt used in GRAPH PLANNING for eliciting LLM's preference over the planning directions in zero-shot.

```
You will be given an undirected graph search problem with a few examples. You will decide which search
    direction is easier to solve for the shortest path from the initial to the goal.

** Current problem **
Node 0 is connected to nodes 7, 10
Node 1 is connected to nodes 4, 5, 11
Node 2 is connected to nodes 8
Node 3 is connected to nodes 6, 11
Node 4 is connected to nodes 1, 6, 9
Node 5 is connected to nodes 1, 6, 7, 11
Node 6 is connected to nodes 3, 4, 5
Node 7 is connected to nodes 0, 5, 9
Node 8 is connected to nodes 2
Node 9 is connected to nodes 4, 7, 10
Node 10 is connected to nodes 0, 9
Node 11 is connected to nodes 1, 3, 5

Initial: 0  Goal: 3

If there is a bottleneck (nodes with few edges connected) at one end of the graph, then it is easier to solve
    for the shortest path from that end. Which direction (forward, from the initial, or backward, from the
    goal) has the bottleneck? Summarize your reasoning in a short paragraph without going through all the
    nodes, and finish your answer with 'Direction with bottleneck: <forward/backward>'.
```

## A.2 ARRAY TRANSFORMATION

**Configurations.** There are six possible functions used: `repeat`, `cut`, `shift_left`, `shift_right`, `reverse`, and `swap`. Depending on the set of functions used (e.g., {`repeat`, `cut`, `shift_left`, `shift_right`}), we first sample an random array of size 4 as the initial array, sample a random set of three functions, and then apply these functions to the initial array to get the goal one — if `cut` is sampled, we then first invert all the functions, reverse the order, apply the functions, and then reverse the initial and goal arrays. We limit that `repeat` can appear only once among the three to ensure the goal array is not too long.

**Prompts.** The prompts used for sampling candidate solutions and self-verifying them are shown in listing 5, listing 6, and listing 7. listing 8 shows the prompt used for eliciting LLM's preference

over the planning directions in zero-shot. We use temperature $T = 0$ for self-verifying, eliciting preference, and the first planning attempt, and $T = 0.5$ for the rest of planning attempts.

Listing 5: Header used in the prompt for ARRAY TRANSFORMATION. The functions shown depend on the set of functions used.

```
A random sequence of three of the below functions transform the initial array into the final array.
Given initial and final, output the sequence of transformations.

def reverse(x):
  # reverse the sequence
  return x[::-1]

def shift_left(x):
  # shift the sequence to the left by one
  return x[1:] + x[:1]

def shift_right(x):
  # shift the sequence to the right by one
  return [x[-1]] + x[:-1]

def swap(x):
  # swap the first and last elements
  return x[-1:] + x[1:-1] + x[0:1]

def repeat(x):
  # repeat the sequence once
  return x + x

def cut(x):
  # cut the sequence in half
  assert x[:len(x) // 2] == x[len(x) // 2:]
  return x[:len(x) // 2]
```

Listing 6: Sample prompt used in ARRAY TRANSFORMATION for planning candidate solutions, excluding the header.

```
***** Examples:
Initial: [4, 3, 7, 4, 4, 3, 7, 4]
Final: [7, 4, 4, 3]
Initial to Final Steps:
  cut: [4, 3, 7, 4]
  shift_right: [4, 4, 3, 7]
  shift_right: [7, 4, 4, 3]
Functions: [cut, shift_right, shift_right]

Initial: [8, 0, 0, 2]
Final: [0, 2, 8, 0, 0, 2, 8, 0]
Initial to Final Steps:
  shift_left: [0, 0, 2, 8]
  repeat: [0, 0, 2, 8, 0, 0, 2, 8]
  shift_left: [0, 2, 8, 0, 0, 2, 8, 0]
Functions: [shift_left, repeat, shift_left]

Initial: [2, 9, 6, 5]
Final: [6, 5, 2, 9, 6, 5, 2, 9]
Initial to Final Steps:
  shift_right: [5, 2, 9, 6]
  shift_right: [6, 5, 2, 9]
  repeat: [6, 5, 2, 9, 6, 5, 2, 9]
Functions: [shift_right, shift_right, repeat]

***** Current problem:
Initial: [4, 0, 0, 5]
Final: [0, 5, 4, 0, 0, 5, 4, 0]
Please solve with the exact same format. Do not repeat the problem.
```

Listing 7: Sample prompt used in ARRAY TRANSFORMATION for self-verifying the candidate solutions, excluding the header.

```
***** Examples:
Initial: [3, 0, 8, 8, 3, 0, 8, 8]
Desired Final: [8, 8, 0, 3]
Functions: [shift_left, cut, shift_right]
Verify initial to final steps:
  shift_left: [3, 0, 8, 8, 3, 0, 8, 8][1:] + [3, 0, 8, 8, 3, 0, 8, 8][:1] -> [0, 8, 8, 3, 0, 8, 8, 3]
  cut: [0, 8, 8, 3, 0, 8, 8, 3] half -> [0, 8, 8, 3] and [0, 8, 8, 3] equal -> [0, 8, 8, 3]
  shift_right: [[0, 8, 8, 3][-1]] + [0, 8, 8, 3][:-1] -> [3, 0, 8, 8]
  actual final: [3, 0, 8, 8], desired final: [8, 8, 0, 3], does not match
  Incorrect

Initial: [1, 5, 7, 2]
Desired Final: [7, 2, 1, 5, 7, 2, 1, 5]
Functions: [shift_right, shift_right, repeat]
Verify initial to final steps:
```

```
shift_right: [[1, 5, 7, 2][-1]] + [1, 5, 7, 2][:-1] -> [2, 1, 5, 7]
shift_right: [[2, 1, 5, 7][-1]] + [2, 1, 5, 7][:-1] -> [7, 2, 1, 5]
repeat: [7, 2, 1, 5] + [7, 2, 1, 5] -> [7, 2, 1, 5, 7, 2, 1, 5]
actual final: [7, 2, 1, 5, 7, 2, 1, 5], desired final: [7, 2, 1, 5, 7, 2, 1, 5], match
Correct

Initial: [5, 5, 0, 2, 5, 5, 3, 2]
Desired Final: [4, 2, 0, 5, 5]
Functions: [shift_left, cut, shift_right]
Verify initial to final steps:
  shift_left: [5, 5, 0, 2, 5, 5, 3, 2][1:] + [5, 5, 0, 2, 5, 5, 3, 2][:1] -> [5, 0, 2, 5, 5, 3, 2, 5]
  cut: [5, 0, 2, 5, 5, 3, 2, 5] half -> [5, 0, 2, 5] and [5, 3, 2, 5] not equal -> cut failed
  Incorrect

Initial: [2, 5, 9, 5]
Desired Final: [2, 5, 9, 5, 2, 5, 9, 5]
Functions: [shift_right, repeat, shift_left]
Verify initial to final steps:
  shift_right: [[2, 5, 9, 5][-1]] + [2, 5, 9, 5][:-1] -> [5, 2, 5, 9]
  repeat: [5, 2, 5, 9] + [5, 2, 5, 9] -> [5, 2, 5, 9, 5, 2, 5, 9]
  shift_left: [5, 2, 5, 9, 5, 2, 5, 9][1:] + [5, 2, 5, 9, 5, 2, 5, 9][:1] -> [2, 5, 9, 5, 2, 5, 9, 5]
  actual final: [2, 5, 9, 5, 2, 5, 9, 5], desired final: [2, 5, 9, 5, 2, 5, 9, 5], match
  Correct

***** Current problem:
Initial: [4, 0, 0, 5]
Final: [0, 5, 4, 0, 0, 5, 4, 0]
Functions: [shift_left, repeat, shift_left]
Please verify initial to final steps with the exactly same format. Do not repeat the problem.
```

Listing 8: Sample prompt used in ARRAY TRANSFORMATION for eliciting LLM's preference over the planning directions in zero-shot, excluding the header.

```
***** Current problem:
Initial: [6, 4, 0, 4]
Final: [0, 4, 4, 6]

The problem can be solved either in the forward direction (from initial to final), or by flipping the problem
    first (final becomes initial, initial becomes final) and then solving in the new forward direction.
    Which direction would you like to solve in? Think about possible bottleneck where fewer search steps are
    needed. Summarize your reasoning in a short paragraph without going through the intermediate steps and
    arrays, and finish your answer with 'Direction with bottleneck: <forward/flipped>'.
```

## A.3 BLOCKSWORLD

**Configurations.** We use the problems from the `task_1_plan_generation` task (validity) in the PlanBench benchmark (Valmeekam et al., 2024) without modifying them.

**Prompts.** listing 9 shows the original prompt from the benchmark. However, we find LLM often struggle to plan by following the examples in the original prompt, often mistaking the correct initial state. Instead, we use a two-step approach where LLM first summarizes the initial conditions into short format (e.g., "yellow on red; blue; orange") (listing 10), and then plan — during planning, LLM generates intermediate states in short form to help it reason the next steps (listing 11). listing 12 shows the prompt used for eliciting LLM's preference over the planning directions in zero-shot. We use temperature $T = 0$ for self-verifying, eliciting preference, and the first planning attempt, and $T = 0.4$ for the rest of planning attempts.

Listing 9: Original prompt (header, example, and current problem) from the PlanBench benchmark used in BLOCKSWORLD. Notice that the goal state can be partial.

```
You will play with a set of blocks where you need to arrange the blocks into stacks.

[POSSIBLE ACTIONS]
Pick up a block
Unstack a block from on top of another block
Put down a block
Stack a block on top of another block

[RULES]
Only pick up or unstack one block at a time.
Only pick up or unstack a block if hand is empty.
Only pick up a block if the block is on the table and the block is clear. A block is clear if the block has no
    other blocks on top of it and if the block is not picked up.
Only unstack a block from on top of another block if the block being unstacked was really on top of the other
    block.
Only unstack a block from on top of another block if the block being unstacked is clear.
Once you pick up or unstack a block, you are holding the block.
Only put down a block that you are holding.
Only stack a block on top of another block if you are holding the block being stacked.
```

```
Only stack a block on top of another block if the block onto which you are stacking the block is clear.
Once you put down or stack a block, your hand becomes empty.
Once you stack a block on top of a second block, the second block is no longer clear.

[[EXAMPLE]]
As initial conditions I have that, the red block is clear, the blue block is clear, the yellow block is clear,
    the hand is empty, the blue block is on top of the orange block, the red block is on the table, the
    orange block is on the table and the yellow block is on the table.
My goal is to have that the orange block is on top of the blue block.
My plan is as follows:
[PLAN]
unstack the blue block from on top of the orange block
put down the blue block
pick up the orange block
stack the orange block on top of the blue block
[PLAN END]

[STATEMENT]
As initial conditions you have that, the red block is clear, the yellow block is clear, the hand is empty, the
    red block is on top of the blue block, the yellow block is on top of the orange block, the blue block
    is on the table and the orange block is on the table.
Your goal is to have that the orange block is on top of the red block.
```

Listing 10: Sample prompt used in BLOCKSWORLD to summarize the initial and goal state, excluding the header.

```
[[EXAMPLE]]

[STATEMENT]
As initial conditions you have that, the red block is clear, the yellow block is clear, the hand is empty, the
    red block is on top of the blue block, the yellow block is on top of the orange block, the blue block
    is on the table and the orange block is on the table.
Your goal is to have that the orange block is on top of the red block.

First you summarize the init state and goal:

[PLAN]
init state (each clause is a stack): red on blue; yellow on orange
goal: orange on red
[PLAN END]

[[CURRENT PROBLEM]]

[STATEMENT]
As initial conditions you have that, the blue block is clear, the hand is empty, the blue block is on top of
    the orange block, the orange block is on top of the yellow block, the yellow block is on top of the red
    block and the red block is on the table.
Your goal is to have that the red block is on top of the orange block and the yellow block is on top of the
    red block.

Please follow the format and generate the init state and goal for the current problem. Make sure the stacks
    are combined if they should, e.g., 'red on blue; yellow on red' should be combined as 'yellow on red on
    blue'.
```

Listing 11: Sample prompt used in BLOCKSWORLD to plan the steps.

```
[[EXAMPLE]]

[STATEMENT]
init state (each clause is a stack): orange on red on blue; yellow
goal: red on blue; yellow on orange.

Your plan is as follows:

[PLAN]
unstack the orange block from on top of the red block
(orange on hand; red on blue; yellow)
put down the orange block
(red on blue; yellow; orange)
pick up the yellow block
(yellow on hand; red on blue; orange)
stack the yellow block on top of the orange block
(red on blue; yellow on orange) Goal satisfied
[PLAN END]

[[CURRENT PROBLEM]]

[STATEMENT]
init state (each clause is a stack): yellow on red on orange; blue
goal: blue on orange on yellow on red

Please follow the format and generate your plan for the current problem. Start with [PLAN]
```

Listing 12: Sample prompt used in BLOCKSWORLD to elicit LLM's preference over the planning directions in zero-shot, excluding the header.

```
[STATEMENT]
As initial conditions you have that, the red block is clear, the yellow block is clear, the hand is empty, the
    red block is on top of the blue block, the yellow block is on top of the orange block, the blue block
    is on the table and the orange block is on the table.
Your goal is to have that the orange block is on top of the red block.

The problem can be solved either in the forward direction (from initial condition to goal), or by flipping the
    problem first (goal becomes initial, initial becomes goal) and then solving in the new forward
    direction. Which direction would you like to solve in? Think about possible bottleneck where fewer
    search steps are needed. Summarize your reasoning in a short paragraph without going through the
    intermediate steps and states, and finish your answer with 'Direction with bottleneck: <forward/flipped
    >'.
```

## B    APPENDIX - ADDITIONAL RESULTS

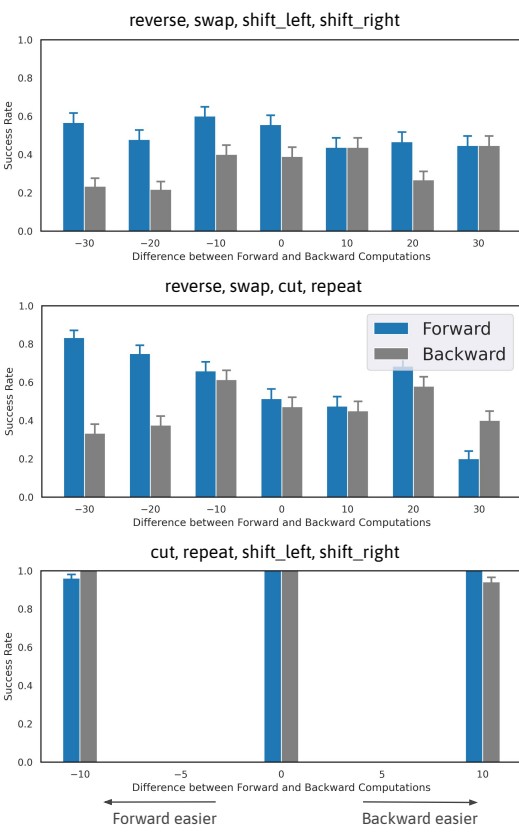

Figure 12: Success rates achieved by forward and backward planning in ARRAY TRANSFORMATION vs. difference between forward and backward BFS computations. In general LLM plans better in the direction of fewer computations needed, but the forward direction outperforms backward. In the last set of experiments, we find Back works well by recognizing arrays that should be repeated or cut, and thus there is no visible difference between Fwd and Back.

