# OpenReview forum: "Thinking Forward and Backward: Effective Backward Planning with Large Language Models"
_ICLR.cc/2025/Conference — Submitted to ICLR 2025_

### Official Review · Reviewer_ceDr · 2024-10-25

**Soundness:** 2
**Presentation:** 2
**Contribution:** 2
**Rating:** 3
**Confidence:** 4

**Summary:**

The paper first investigates the performance of LLMs on a set of planning problems where both forward and backwards planning can be done (3 domains are given in the paper: shortest path in graphs, array and Blocksworld). The paper measures the difficulty of planning in a particular direction with the size of search space (in that direction). Based on this, the authors empirically observe and claim in Sec. 3.3 that 1) LLMs, in general, plan better in the "easier" planning direction and 2) ceteris paribus, LLMs plan worse in the backwards direction as compared to the forward direction. The authors hypothesize that 2) "... may be attributed to the forward autoregressive nature of LLM output generation, as well as biases from the training dataset."

Then, the paper, in Sec. 4, proposes a method which flips the problem backwards and plan forward (while handling some subtleties) for the flipped problem. Combined with self-verification and some sampling, the experiments show that their method outperforms other baselines over the 3 planning domains.

**Strengths:**

1. The paper's approach in developing the main method is, to a certain extent, systematic. It first verifies that some planning problems are easier solved from the backwards direction (due to smaller action spaces) then show that LLMs perform better in the direction which is easier. However, the paper also found that LLMs cannot plan well directly in the backward direction. Combining these two observations, the paper then introduces its method of flipping the backward problem.

2. The experimental results are laid out in a systematic manner, where each section tries to explain a certain statement or claim in the paper.  The experimental setup is also done well, although I have some comments about how the authors tried to link the results with certain claims (see next section).

**Weaknesses:**

I have several concerns regarding how claims in the papers are supported by the experiments. These concerns also impact the novelty and soundness of the authors' main method (**Fwd-Flip**).


1. The authors mentioned that **Fwd** is the standard forward planning method, **Flip** is applying the forward planning method to the flipped problem, and **Fwd-Flip** is "randomly choose either forward direction of the original problem or forward direction of the flipped problem and then plan". Therefore, the only difference between **Fwd-Flip** and **Fwd**/**Flip** is that instead of deciding which direction to plan, the planning direction is randomly selected. However, the experimental results in Table 2 show that **Fwd-Flip** achieves much higher success rate than **Fwd** or **Flip**. To me, this is result is questionable because if **Fwd-Flip** is just selecting **Fwd** or **Flip** randomly, how can it perform much better? In fact, I would expect, with just random sampling, **Fwd-Flip** to perform exactly the average of **Fwd** and **Flip**? I also noticed also that Fwd-Flip is different from what was written in the introduction, which is "Given a planning problem, we ask LLMs to sample possible plans in the forward direction of both the original problem and the flipped one, and then self-verify (Stechly et al., 2024) all the plans before choosing the final one". Could the authors clarify?


2. I also noticed that both **Fwd** and **Flip** (on their own) have similar performances over all planning tasks in Table 2. This implies that on average, flipping the problem has no benefits over the direct forward planning method. While I understand that the authors observed that certain planning directions could be easier in some tasks (due to difference in action spaces), it is clear from the experiment results that a particular planning direction is not inherently better in any of the tasks. Hence, I argue that the paper's main method, **Fwd-Flip** does not actually exploit this observation effectively. I have a suggestion on how the method can be improved: during test-time, we can cleverly decide whether the problem is easier if we flipped it (using some simple heuristics). Then, we can either apply **Fwd** or **Flip** based on our heuristic; this is more sound than **Fwd-Flip** proposed in this paper, which is simply selecting **Fwd** or **Flip** *randomly*. In later experiments, the authors do use the LLM to reason about which planning direction to use, but its effectiveness seems to be restricted to Undirected Graph Planning.


3. Some claims in the papers are briefly stated in words but are not clearly substantiated from the experiments. For example:
* Sec. 4 claims that flipping the problem "... avoids the bias of weak LLM planning in the backward direction." However, Figure 3 shows that for Directed Graphs, for problems with no difference in forward and backward computations, flipping still performs worse than the forward problem. So, it is unclear if flipping the problem entirely avoids the bias.
* There are results such as Figure 3 and Figure 10 that are used to justify the paper's claims that "... LLM plans better in the direction
of fewer computations needed, but the forward direction outperforms backward in general ...". However, they are only shown for one planning task (instead of the 3 domains which the paper claims its focus is on). Since the paper mainly relies on empirical observations to justify its claims, I think the same figure should be replicated for other tasks as well (especially that I noticed there are additional space available in the main paper).
* Table 3 tries to show that making the LLM decide which planning direction to use during test-time could improve performance. But somehow the table does not have the results for **Fwd-Flip**, the author's main proposed method for us to compare **Fwd-Flip-Reason** to. We also cannot just look at the previous results for **Fwd-Flip** because the authors mentioned that they used a different hyperparameter $M$ as compared to that in Table 2.

**Questions:**

1. "We expect our proposed Fwd-Flip to outperform Fwd, Back, Flip, Fwd-Flip." I think the last Fwd-Flip is a typo - it should be Fwd-Back right?
2. The method written in the introduction - "Given a planning problem, we ask LLMs to sample possible plans in the forward direction of both the original problem and the flipped one, and then self-verify (Stechly et al., 2024) all the plans before choosing the final one", is different from what was presented as the main method, which is **Fwd-Flip**. The description of **Fwd-Flip** given in the experiments is "randomly choose either forward direction of the original problem or forward direction of the flipped problem and then plan". This is very confusing because these are two totally different methods. I'd like the authors to clarify which is their main proposed method.

---

### Official Review · Reviewer_LPQo · 2024-11-01

**Soundness:** 1
**Presentation:** 2
**Contribution:** 1
**Rating:** 1
**Confidence:** 5

**Summary:**

This paper investigates the possible usage of LLMs as an automated planner for solving classical planning tasks (deterministic, fully observable). Especially, the paper studies the regression planning approach. The paper also proposes a heuristic approach that switches the initial state and the goal depending on the progress of solving the problem. Overall, experiments were conducted on path finding in graphs, array transformation, and the blocksworld domains.

**Strengths:**

This paper explores regression planning with LLMs. It seems like the authors are not aware of the literature on automated planning and classical AI, from what's written. so it is an attempt to re-invent the known results in the 70s to 90s in the context of large language models.

**Weaknesses:**

This paper mentions bidirectional search in the related work. However, it is mentioned nowhere in the paper about regression planning. "Thinking forward and backward" reminds me of a fancy title "Thinking fast and slow" and the title also shows effective backward planning. However, it is doubtful if that's the case.
If we hold back and remove the LLM portion, then the paper suggests solving planning problems by alternating the initial state and the goals. In general, goals are not a single state as the initial state in most of the problems. It is doubtful how such a switching strategy can lead to systematic improvement as we can always choose one of the goal states as a new initial state in a different problem.
One of the weaknesses here is the lack of coverage of the relevant literature.

There are several misleading statements such as
"classical planning algorithms such as BFS" line 141. ==> BFS is not a classical planning algorithm
"bottleneck" reminds me of the concept of information bottleneck in information theory. However, it is not relevant to this context.
It is not clear whether the "bottleneck effect" is relevant in this context.


The details of the implementation of the planner are vague.
How possible actions are generated? How the next state will be generated given actions?

**Questions:**

Question 1. What are the problem statistics of the three problem domains? They are all classical planning problems so we could see the action schema, and the number of state predicates to have a sense of the difficulty of the problem. What problem instances were tested? How many objects were tried? How the initial states and the goals were determined? It is not difficult to write down PDDL specifications for those problems.

Question 2. Given a state observation, how the next actions are generated? Are the next actions sound and applicable to each state always?
How the next state is generated? How the goal test was done?

Question 3. How the natural language reasoning version of the classical planning problems are obtained, for the graph planning an array transformation? Are they a natural description of the problem? Or is it utilizing artificial patterns?

Question 4. Blocksworld problem would be the only single problem domain that has multiple goal states.
In regression planning, the difficulty is to maintain a set of states instead of a single state if we proceed to perform a search.
The experiment results also confirm such known issues. In the other two domains, regression modes are claimed to perform well. But I think it depends on the distribution of the initial state and the goal state, not the goal states here. I think if we simply flip the initial state and the goal state, we will obtain the opposite results. Could you explain this?

---

### Official Review · Reviewer_d5hd · 2024-11-03

**Soundness:** 2
**Presentation:** 3
**Contribution:** 2
**Rating:** 5
**Confidence:** 4

**Summary:**

This paper uses LLMs for planning.  In particular it looks at the difference between planning forward (from initial state to the goal) vs planning backward (from the goal to the initial state), and notes that backward planning may work better for some problems.  For problems where backward planning is more efficient, the paper introduces a method for flipping the problem so that forward planning can be used from the goal to the initial state.  This leads to the question: Can LLMs plan better if they reason in the backward direction?”

**Strengths:**

The idea of flipping the problem has merit for some problems.  With more careful consideration of the general claims about the capabilities of LLMs (many of which actually seem problem dependent) and a more clear definition of which problems this will work for and which it will not, and why (including a discussion of what makes some actions non-invertible), this would be a strong paper.

**Weaknesses:**

Overall: Many general claims are made about the capabilities of LLMs (e.g. they exhibit a systematic bias towards forward planning), and these claims are backed by experiments.  However, it's not clear that the experimental results justify these broad claims.  Many of the experimental results seem very domain-dependent.  For example, Directed Graph Planning has a built in directional bias.

1.	The paper uses the term “backward planning” but all the planning is actually in the forward direction, it’s just that sometimes they “flip” the problem (making the old goal the new initial state and the old initial state the new goal).
2.  The method proposed for flipping the problem assumes that all actions can be inverted, and it is not clear that this is always the case (so the applicability of “flipping” will depend on whether or not the actions in the domain are, in fact, invertible).
3.	Regarding the claims that “LLM exhibit a systematic bias of performing worse when planning backwards” may be attributed to the forward auto-regressive nature of LLM output generation or training data biases:  There is not enough evidence to support this as a general claim.  Experiments were conducted with only three domains, and at least one (Graph Planning) has by its very nature a built in bias in the forward direction.
4.	Figure 2 as an algorithmic description is not clear, and does not seem to match up with the text in lines 168-17.
5.	The general conclusions about LLMs drawn from the data presented in Figure 3 (LLMs plan better in the forward direction, and that flip solves this bias) seem suspect given that the directed case in this domain has a built in bias to forward planning so it’s not surprising at all that flip would do well in this case.
6.	Also, regarding Figure 3; most of the cases where flip outperforms forward planning are also cases where backward planning already also outperformed forward planning.

**Questions:**

1.   What is it about some actions that make them non-invertible?  How does this affect your proposed method (which assumes that all actions are invertible)?
2.   Is it important that inverse actions are actually feasible actions that respect physics and make sense?  (e.g. if an action is to “drill a hole”, you can’t really un-drill a hole; does this even matter?).
3.	Bi-directional search is mentioned in related work, but it’s not mentioned at all in the text.  The approaches that involve both forward and backward planning seem very similar to bi-directional search and may warrant more discussion in the paper.  Are there lessons from bi-directional search that are relevant here?
4.	Doesn’t the Graph Planning domain have built in bias for directed graphs?  Meaning they may only be solvable in one direction?
5.	In the description of how planning problems are flipped; the authors assume that every action has an inverse action.  It’s not clear that this would always be true.  Could you please include discussion of this?  Could you give examples where there is not an inverse action?  (e.g. if an action is to “drill a hole”, you can’t really un-drill a hole can you)?

---

### Official Review · Reviewer_SVvX · 2024-11-05

**Soundness:** 2
**Presentation:** 2
**Contribution:** 2
**Rating:** 3
**Confidence:** 4

**Summary:**

The authors propose a method that uses large language models (LLMs) for planning by “flipping” the problem. In this approach, the initial and goal states are reversed, a plan is generated by the LLM, and then this plan is reversed to solve the original problem. The authors demonstrate that this method can enhance the planning performance of LLMs.

**Strengths:**

The paper conducts a thorough set of experiments to test whether flipping the initial and goal states of problems improves LLM performance. The authors also examine whether the LLM can reason about whether or not to flip the problem.

**Weaknesses:**

The impact of the proposed approach is somewhat unclear. For example, prior research [1] has shown that LLM planning performance significantly deteriorates when the syntax of the planning domain is obfuscated, even when the causal structure remains intact (referred to as “Mystery” domains in [1]). It would have been insightful to see how this approach performs in such obfuscated domains. This also suggests that the computational complexity of the underlying domain doesn’t necessarily affect LLM performance. Therefore, flipping the problem’s initial and goal states to reduce computational complexity might not necessarily benefit LLMs, as they may not be sensitive to these complexity reductions in the same way formal planners are.

Based on my understanding, the method described in Section 4 (always flipping the original problem) aligns with the “Flip” approach in Section 5. Flip performs worse than “Fwd” planning, contrary to what is suggested in Section 5. The proposed approach mentioned in Section 5 appears to be “FwdFlip,” where the decision to flip is made at random. It would be helpful to clarify the exact nature of the proposed approach. Additionally, it would be valuable to see details within the FwdFlip method, such as the success rates within Fwd and Flip respectively.

**Questions:**

1. Are the results of Flip related methods based on the solution to the flipped problem or the original problem? If it is w.r.t the original problem, how are the errors distributed between the solution of the flipped problem being wrong and the flipping of the plan being wrong?

2. Is the flipped initial state in Blocksworld verified given that the LLM is generating a full state from the partial goal state?

---

### Meta-Review · Area_Chair_9qPn · 2024-12-22

**Metareview:**

The paper explores the use of large language models for classical planning tasks. The authors propose a method where planning problems are "flipped" by reversing the initial and goal states. This approach, combined with self-verification and sampling, aims to improve planning performance by addressing perceived limitations of LLMs in backward planning. The authors conduct experiments across three planning domains: graph pathfinding, array transformations, and Blocksworld.

While this paper attempts to identify the challenges of backward planning in LLMs and proposes an intuitive solution (flipping problems) based on empirical observations, the reviewers raised several concerns:

- R1, R4 noted inconsistencies in the description of the proposed method.
- R3 pointed out that the method largely resembles classical planning techniques. R2, R4 also noted the absence of bi-directional search discussions, which are closely related to the proposed approach.
- Several general claims are not well-supported by experimental results (R2, R4).
- R4 raised concerns about experimental validity.
- The paper assumes invertibility of actions, which is a critical limitation (R2, R3).

**Additional Comments On Reviewer Discussion:**

No rebuttal is provided.

---

### Decision · Program_Chairs · 2025-01-22

Reject